# A Single Dose of ChAdOx1 nCoV-19 Vaccine Elicits High Antibody Responses in Individuals with Prior SARS-CoV-2 Infection Comparable to That of Two-Dose-Vaccinated, SARS-CoV-2-Infection-Naïve Individuals: A Longitudinal Study in Ethiopian Health Workers

**DOI:** 10.3390/vaccines10060859

**Published:** 2022-05-27

**Authors:** Tesfaye Gelanew, Andargachew Mulu, Markos Abebe, Timothy A. Bates, Liya Wassie, Mekonnen Teferi, Dessalegn Fentahun, Aynalem Alemu, Frehiwot Tamiru, Gebeyehu Assefa, Abebe Genetu Bayih, Fikadu G. Tafesse, Adane Mihret, Alemseged Abdissa

**Affiliations:** 1Armauer Hansen Research Institute, Jimma Road, ALERT Campus, Addis Ababa P.O. Box 1005, Ethiopia; andargachew.mulu@ahri.gov.et (A.M.); markos.abebe@ahri.gov.et (M.A.); liya.wassie@ahri.gov.et (L.W.); mekonnen.teferi@ahri.gov.et (M.T.); dessalegnfemu@gmail.com (D.F.); aynalema782@gmail.com (A.A.); frehwottamiru@gmail.com (F.T.); gebeyehu03@yahoo.com (G.A.); abebe.genetu@ahri.gov.et (A.G.B.); adane.mihret@ahri.gov.et (A.M.); alemseged.abdissa@ahri.gov.et (A.A.); 2Department of Molecular Microbiology & Immunology, Oregon Health & Sciences University (OHSU), Portland, OR 97239, USA; bates@ohsu.edu (T.A.B.); tafesse@ohsu.edu (F.G.T.)

**Keywords:** ChAdOx1 nCoV-19, SARS-CoV-2, vaccine, dose, RBD, naïve, prior infection

## Abstract

Single-dose COVID-19 vaccines, mostly mRNA-based vaccines, are shown to induce robust antibody responses in individuals who were previously infected with SARS-CoV-2, suggesting the sufficiency of a single dose for those individuals in countries with limited vaccine supply. However, these important data are limited to developed nations. We conducted a prospective longitudinal study among Ethiopian healthcare workers who received a ChAdOx1 nCoV-19 vaccine. We compared the geometric mean titers (GMTs) of the SARS-CoV-2 receptor-binding domain (RBD)-specific IgG antibodies in 39 SARS-CoV-2 naïve participants and 24 participants previously infected with SARS-CoV-2 (P.I.), who received two doses of ChAdOx1 nCoV-19 vaccine across the two post-vaccination time points (at 8 to 12 weeks post single dose and two dose vaccinations). We noted that the GMT (1632.16) in naïve participants at 8–12 weeks post first dose were comparable to the GMT (1674.94) observed in P.I. participants prior to vaccination. Interestingly, P.I. participants had significantly higher antibody titers compared to naïve participants, after both the first (GMT, 4913.50 vs. 1632.16) and second doses (GMT, 9804.60 vs. 6607.30). Taken together, our findings show that a single ChAdOx1 nCoV-19 dose in previously SARS-CoV-2 infected individuals elicits similar, if not higher, antibody responses to those of two-dose-vaccinated naïve individuals.

## 1. Introduction

Due to a wide variety of viral strains belonging to the severe acute respiratory syndrome coronavirus 2 (SARS-CoV-2) type, coronavirus disease 2019 (COVID-19) causes a broad spectrum of diseases primarily affecting the respiratory and vascular systems of the human body. The SARS-CoV-2 viral strains can be highly transmittable, pathogenic, or capable of evading the adaptive immune responses. SARS-CoV-2 human-to-human transmission occurs mainly through exposure to respiratory fluids carrying infectious virus.

COVID-19 continues to be a major public health concern, causing severe illness and deaths in Ethiopia and the rest of the world alike. Mass vaccination against SARS-CoV-2 is the most effective public health intervention to protect against morbidity and mortality related to SARS-CoV-2 infection [1]. Safe, efficacious, and licensed COVID-19 vaccines, including ChAdOx1 nCoV-19 (AZD1222; Oxford–AstraZeneca), are available [2,3,4,5], despite being challenged by the recurrent emergence of new SARS-CoV-2 variants. Real-world vaccine efficacy studies from developed countries have shown that the current vaccines are able to generate effective humoral and cellular immunity, albeit differential responses are observed between vaccine-induced immunity and hybrid (vaccine-induced immunity combined with natural infection) immunity [6,7]. Several correlates of protection studies have demonstrated that higher antibody titers are associated with decreased risk of subsequent symptomatic SARS-CoV-2 infection [8,9,10,11,12], and several studies from developed countries have revealed the rapid waning of antibody levels among SARS-CoV-2-infection-naïve vaccine recipients compared to those individuals with hybrid immunity [7,13,14,15,16]. Despite the importance of immune durability data for guiding national vaccination strategies, there is a dearth of studies from Ethiopia and other African countries looking at more locally relevant populations.

The ChAdOx1 nCoV-19 vaccine utilizes a replication-deficient adenoviral vector that induces expression of SARS-CoV-2 spike (S) protein in host cells, particularly in the skeletal muscle [17]. Vaccinated individuals subsequently generate antibodies against the spike protein, including those that target the receptor-binding domain (RBD), which contains many neutralizing epitopes. However, they do not generate antibodies against other SARS-CoV-2 structural and non-structural proteins, such as nucleocapsid (N) [18]. Studies have shown a strong correlation between anti-RBD IgG titers and SARS-CoV-2 neutralizing titers [15]. Therefore, in resource-limited countries, it is advantageous to use anti-RBD IgG testing as a proxy for virus neutralization to assess the protection offered by the ChAdOx1 nCoV-19 vaccine. 

As part of the strategy to evaluate the Ethiopian national COVID-19 response through vaccination, we established a longitudinal cohort of healthcare professionals working at the Armauer Hansen Research Institute (AHRI), with and without evidence of prior SARS-CoV-2 infection and determined their levels of ChAdOx1 nCoV-19 vaccine-induced anti-RBD IgG titers across two-time points. The present study generated evidence of the durations of ChAdOx1 nCoV-19 vaccine-induced humoral responses and the long-term effects of prior SARS-CoV-2 infection on subsequent vaccine-induced responses.

## 2. Materials and Methods

### 2.1. Study Design and Participants

We conducted a longitudinal prospective study constituting healthcare professionals from AHRI, who were also one of the priority target recipients of the ChAdOx1 nCoV-19 vaccine. Vaccination was offered through the Ethiopian Ministry of Health’s national COVID-19 vaccination campaign. In this analysis, only participants who were vaccinated with the ChAdOx1 nCoV-19 vaccine were included. About 5 mL of venous blood was collected from a total of 63 (*n* = 27 female and *n* = 36 male) participants, aged 24–59 years (mean of 38.1 ± standard deviation of 8.36) before vaccination and in two subsequent post-vaccination follow-ups (Figure 1) to monitor humoral responses to the vaccine. The first (hereafter designated as 1d + 8–12w) was conducted at 8–12-weeks after the first dose, whereas the second follow-up (designated hereafter as 2d + 8–12w) was done 8–12 weeks after the second dose (Figure 1). All participants received their first dose between 23 March 2021 and 31 March 2021. The interval between the first and the second doses was 12 weeks (range 83–97 days). Participants were then stratified into two groups based on previous exposure to SARS-CoV-2 before vaccination and hereafter denoted as naïve and P.I. (Figure 1). The two groups, naïve (*n* = 39) and P.I. (*n* = 24), contained similar distributions of age and sex (Figure 1 and Table 1). Table 1 summarizes the demographics of each group of study participants. In addition, participants completed a questionnaire at each visit regarding their history of RT-PCR confirmed SARS-CoV-2 infection.

### 2.2. ELISA Methods 

Prior to analyzing the presence of anti-RBD IgG antibodies, each serum sample was treated with Triton X-100 at the final concentration of 0.1% and incubated at room temperature (RT) for 30 min. This procedure was performed to reduce risk from any potential virus in serum [19]. Detection of anti-RBD IgG antibodies in the sera was done using a validated in-house ELISA, as described previously [20]. To determine the end titer in seropositive serum samples, a two-fold serial dilution starting at 1: 200 in a 96-well ELISA plate was done. The end titer was defined as a serum dilution at which the observed optical density (OD) at 450 nm reads matched that of the OD450 readout for a pre-COVID-19 serum sample diluted at 1:200 and included in each ELISA run as a negative control. Inter-assay variability was normalized by including a convalescent serum in each run of titration ELISA.

To rule out the influence of SARS-CoV-2 breakthrough infections on vaccine-induced antibody response, all post-vaccine sera of the cohort were also analyzed in an in-house-validated ELISA for the presence of measurable antibodies specific to a recombinant SARS-CoV-2 nucleocapsid protein (NP) (BEI Resources). In addition, participants were asked through a survey if they had had a positive PCR test in the past after vaccination. 

### 2.3. Statistical Analysis

GraphPad Prism version 8.0 for Windows (GraphPad Software, La Jolla California USA) was used for statistical analyses. We measured anti-RBD IgG end titers ranging from 1:200 to 1:51200. Graphs were plotted using log10-transformed anti-RBD IgG end titer values. Table 1 comprises the calculated geometric means (GM) titers. As some participants dropped out, particularly following the second dose, the number of blood samples provided after the first dose varied among participants enrolled at baseline/prevaccination (Figure 1 and Table 1). Hence, we used an unpaired non-parametrical Mann–Whitney test to compare the mean differences in log10 anti-RBD IgG titers between time points of serum collection and to examine the influences of age and sex on the anti-RBD antibody responses in our small cohort. A *p*-value < 0.05 was considered statistically significant, and *p* >0.05 was considered non-significant (ns). Participants who did not provide any post-vaccine samples were excluded from the analysis.

## 3. Results

### 3.1. Post-Vaccination Seropositivity

We assessed the seropositivity to anti-RBD IgG antibodies after the first dose and second dose of the ChAdOx1 nCoV-19 vaccine among participants who provided a blood sample for antibody testing at each post-vaccination follow-up. At 8–12 weeks post first dose, detectable anti-RBD IgG antibodies were found in 22 (64.7%; 22/34) naïve and 21 P.I. (87.5%, 21/23) participants (Figure 2B). Notably, three P.I. participants who had measurable anti-RBD IgG antibodies prior to first dose administration underwent seroreversion at 8–12-weeks post first dose. At 8–12 weeks post second dose, anti-RBD IgG antibodies were detected in serum samples from all the 32 (16 naïve and 16 P.I.) participants that provided blood samples. 

All post-vaccination cohort sera were tested in an in-house NP-based indirect ELISA to rule out the impact of post-vaccination asymptomatic breakthrough infection on the vaccine-induced anti-RBD antibody response. We detected anti-NP IgG antibodies in only two SARS-CoV-2 naïve vaccine recipients at 8–12 weeks post single dose vaccination follow-up, and those anti-NP seropositive participants were excluded from the analysis. 

### 3.2. Comparison of Anti-RBD IgG Antibody Titers

We noted that the GMT (1632.16) of anti-RBD IgG in naïve participants at 8–12 weeks post-first dose was comparable to the GMT (1674.94) observed in P.I. participants prior to vaccination. Interestingly, P.I. participants had significantly higher antibody titers compared to naïve participants, after both the first (GMT, 4913.50 vs. 1632.16) and second doses (GMT, 9804.60 vs. 6607.30) (Table 1). 

As shown in Figure 2A, we observed a significant (*p*-value < 0.05) increase in the mean titer of log10 anti-RBD IgG antibodies in naïve participants 8–12 weeks post-second dose compared to the level of the mean titer of log10 anti-RBD IgG antibodies that was observed 8–12-weeks post-first dose. On contrary, we did not observe a statistically significant (*p*-value > 0.5) difference in the level of mean long10 anti-RBD IgG titer in P.I. participants compared to those observed post single dose vaccination (Figure 2B). 

Figure 3 compares the mean log10 anti-RBD IgG antibody tiers for different age groups and sexes at the two post-vaccination time points. We observed no statistically significant difference (*p*-value > 0.05) in the mean log10 anti-RBD IgG titers between age groups 21–40 and 40–59 (Figure 3A,B) or between males and females (Figure 3C,D), irrespective of participants being naïve or P.I., although these observations are limited by the number of participants in the studied subcategories in the cohort

## 4. Discussion

Immune protection following either vaccination or natural infection with SARS-CoV-2 decreases overtime [21]. Although the minimum antibody titer that correlates with protection has not yet been established, a decreased antibody titer is shown to be associated with an increased risk of subsequent symptomatic SARS-CoV-2 infection [8,10,11,12]. In the present study, immunologically naïve participants had relatively low average mean titer anti-SARS-CoV-2 RBD IgG responses at 8–12 weeks post first dose of ChAdOx1 nCoV-19, comparable with those of pre-vaccinated participants with previous exposure to SARS-CoV-2. This finding is consistent with other studies on mRNA-based vaccines [7,13]. As reported elsewhere [22], we also noted the absence of measurable anti-RBD IgG antibodies in 12 of our SARS-CoV-2-infection-naïve participants at three months post single dose. This is consistent with the suggestion made by the UK Joint Committee on Vaccination and Immunization (JCVI): protective immunity elicited by the first dose ChAdOx1 nCoV-19 vaccine will likely last for a duration of 12 weeks [23]. Unexpectedly, we also noted seroreversion in 3 male P.I. participants who had higher prevaccination anti-RBD IgG antibody titers, indicating evidence of a rare event of rapid waning of humoral response in single-dose vaccinated individuals with prior SARS-CoV-2 infection. This could be due to critical medical conditions such as immunosuppression, though in our study, such medical conditions were not systematically recorded. Similarly, seroreversion was observed in HIV-positive patients after receiving two hepatitis A vaccine doses [24].

As expected, individuals who naturally contracted SARS-CoV-2 prior to vaccination had significantly higher anti-RBD IgG antibody titers than immunologically naïve individuals, both after the first and second doses of ChAdOx1 vaccine administration. The vaccine-induced anti-RBD IgG antibody titers produced by participants with prior SARS-CoV-2 infection at three months after a single dose vaccination appeared to be comparable to the antibody titers levels seen after a two-doses for infection-naïve participants. Similar findings have been previously reported for the ChAdOx1 nCoV-19 [25] and BNT162b2 vaccines [7,14,15]. Similarly, SARS-CoV-2-immunologically-naïve individuals developed relatively higher and more durable anti-RBD IgG antibody titers after the second dose of vaccination. Our findings, along with other studies [6,15,26,27], suggest a single vaccine dose in previously infected individuals elicits a similar antibody response to that of double dose vaccination. 

In the present study, participants with anti-SARS-CoV-2 antibodies before the first vaccine injection, regardless of their sex and age (ranged 21–59), developed strong anti-RBD IgG antibodies to the COVID-19 vaccine, and there was no statistically significant variability between the first and the second dose. Given the age range (21–59 years with an average of 38.1 years) of our participants, this finding is not surprising, despite being constrained by the limited sample size; however, an age-dependent decreasing pattern of anti-RBD IgG antibodies titers was reported for similar age groups to our study [28]. On the other hand, the absence of a statistically significant antibody titer difference between male and female participants is surprising and inconsistent with previous reports [26,29], yet agrees with the findings reported by Wheeler et al. [30] and Lee et al. [31].

Regarding side effects of the ChAdOx1 nCoV-19 vaccine, we noted relatively higher local pain at the site of injection and fever within the period of 48 h following the first dose administration of than following the second dose (data not shown). In both cases, these adverse effects had been shown to resolve before 48 h, suggesting the vaccine is tolerable.

A prior study conducted by Harervall et al. [32] among Swedish healthcare workers revealed that administration of a single dose of ChAdOx1 nCoV-19 vaccine could induce equal or greater immune responses in individuals with prior SARS-CoV-2 infection to two doses. Additional studies from different geographical locations, races, and lifestyles are crucial to a better understanding of ChAdOx1 nCoV-19 vaccine-induced immunity. To the best of our knowledge, our study is the first from Ethiopia, if not from the continent of Africa, which provides a possible rationale for a single-dose vaccine regimen in countries with limited access to vaccines. Our findings from Ethiopia, along with evidence from a Western country, Sweden, support the superiority of “hybrid immunity” in eliciting a strong immunity against SARS-CoV-2 infection, comparable to that of two doses of the ChAdOx1 nCoV-19 vaccine in infection-naïve individuals. Recent studies have demonstrated that sera from mRNA-vaccinated individuals with prior infection provide broader cross-neutralizing antibodies against several SARS-CoV-2 variants, including delta variants [33]. Consistent with this, another recent study revealed the mechanism by which hybrid immunity improves B cells and antibodies against SARS-CoV-2 variants [34]. However, a study revealed that the anti-spike antibody titers of BNT162b2 recipients were remarkably higher than those of ChAdOx1 nCoV-19 recipients [13]. Additionally, recent studies showed an increased risk of SARS-CoV-2 omicron infection in both vaccinated and previously infected individuals through evasion of vaccine or infection-induced immune response [35,36], and suggested the need for the rapidly developing new, omicron-specific vaccines and subsequent booster doses. It also remains to be determined if the hybrid immunity elicited by the ChAdOx1 nCoV-19 vaccine will effectively protect the vaccinated individuals from subsequent infections with different SARS-CoV-2 spike variants. Thus, further studies that aim to find a correlation between the level of ChAdOx1 nCoV-19 or any vaccine-induced antibody titer and clinical outcome are required.

## 5. Conclusions

Albeit our study size was small, we noted that individuals who naturally contracted SARS-CoV-2 prior to vaccination had significantly higher anti-RBD IgG antibody titers than immunologically naïve individuals, after both the first and second doses of ChAdOx1 nCoV19 vaccine administration. Generally, policy decisions regarding the dosing regimens consider the current prevalence of SARS-CoV-2 infection, which variants of concern are emerging, population susceptibility, and vaccine supply. Our findings would prompt considerations to prioritize the second dose for SARS-CoV-2-infection-naïve individuals in countries such as Ethiopia, with limited access to the vaccine. However, our findings should not be translated to the omicron variant, as our study was not designed to determine the effectiveness of vaccine-elicited antibodies against omicron. Thus, further studies that compare neutralization of omicron by sera of 1-dose vaccinated individuals with prior infection versus 2-dose vaccinated or 3-dose vaccinated SARS-CoV-2-infection-naïve individuals and vaccinated individuals with omicron breakthrough infection are required to assess the need for a third booster dose over an omicron-specific booster dose in Ethiopia, especially following the omicron pandemic.

## Figures and Tables

**Figure 1 vaccines-10-00859-f001:**
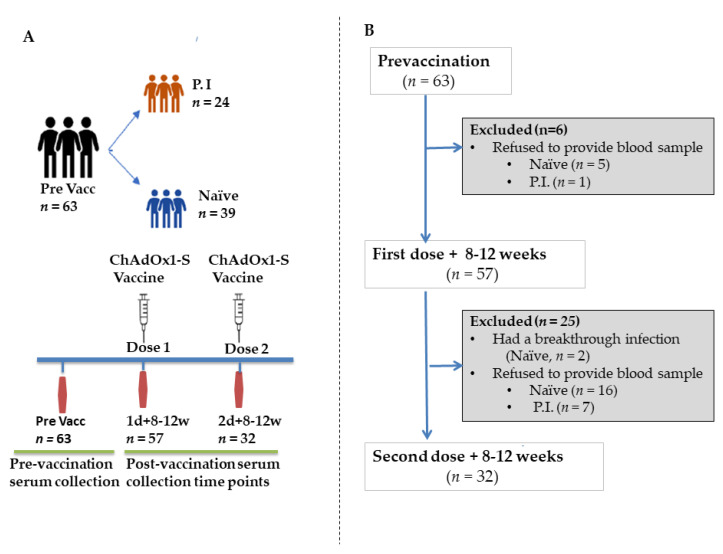
Cohort study design showing the ChAdOx1 nCoV-19 vaccine vaccination regimes and serum sample collection time points: (**A**) Study design with a timeline for vaccination and blood sample collection. (**B**) Schematic representation of the number of participants enrolled at prevaccination and two post-vaccination time points. The first dose and second dose were given to participants prior to the two-time points for blood sample collection. Light red characters (Naïve) represent participants immunologically naïve to SARS-CoV-2 and blue characters (P.I.) represent participants with likely previous SARS-CoV-2 infection. Pre Vacc = baseline or prevaccination; 1d + 8–12w= 8–12 weeks after the first dose; 2d + 8–12w = 8–12 weeks after the second dose.

**Figure 2 vaccines-10-00859-f002:**
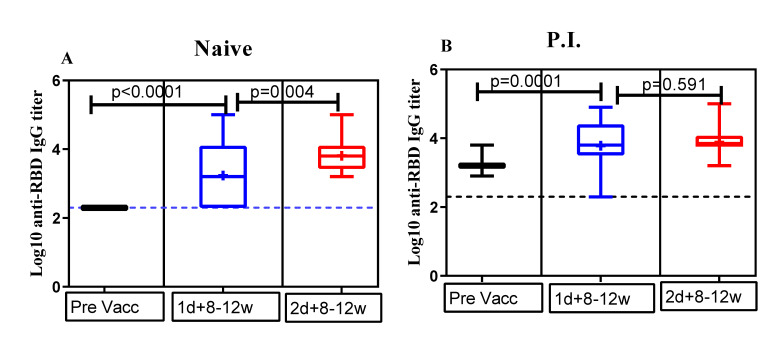
Analysis of the ChAdOx1 nCoV-19 vaccine-induced antibody response in naïve and previously infected (P.I.) participants profile at different time points. Comparison of mean log10 anti-RBD IgG titers between naïve (**A**) and P.I. (**B**) participants across different time points before and after vaccination. Horizontal bars represent the means with 95% CIs of anti-RBD IgG titer levels (transformed to Log10 value) within the indicated groups. The broken line denotes the assay detection limit. Unpaired and non-parametrical Mann–Whitney test was used to compare the mean differences in log10 anti-RBD IgG titers for each time point of serum collection. A *p*-value <0.05 was considered statistically significant, and *p* >0.05 was considered non-significant (ns). Pre Vacc = baseline or prevaccination; 1d + 8–12w = 8–12 weeks after the first dose; 2d + 8–12w = 8–12 weeks after the second dose.

**Figure 3 vaccines-10-00859-f003:**
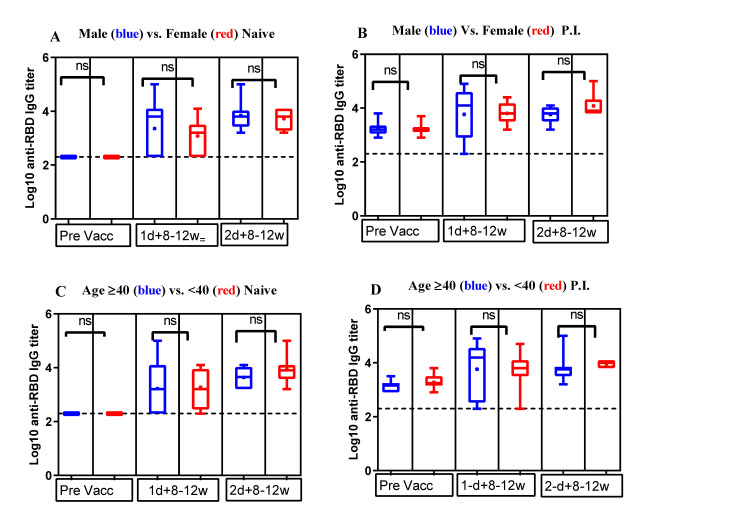
Antibody response by age and sex of naïve and previously infected (P.I.) participants following vaccination. Antibody response comparison of (**A**) naïve and (**B**) P.I. participants by age: 40–59 (blue dots) versus 21–39 years (red dots); antibody response comparison of (**C**) naïve and (**D**) P.I. participants by sex: male (blue dots) versus female (red dots). Pre Vacc = baseline or prevaccination; 1d + 8–12w = 8–12 weeks after the first dose; 2d + 8–12w = 8–12 weeks after the second dose. Unpaired and non-parametrical Mann–Whitney test was used to compare the mean differences in log10 anti-RBD IgG-antibodies titers at each time point of serum collection. A *p*-value <0.05 was considered statistically significant, and *p* > 0.05 was considered non-significant (ns).

**Table 1 vaccines-10-00859-t001:** Baseline characteristics and serostatus of study participants (*n* = 63) and longitudinal humoral immune response.

Character	Subcategory	N ^a^	% ^b^	GMT ^c^
Age in years	≥40	26	41.3	
	<40	37	58.7	
Sex	Male (m ^d^)	36	57.1	
	Female (f ^e^)	27	42.9	
Naïve^f^	Pre Vacc ^h^	39 (23 m,16 f)		200
	1d + 8–12w ^i^	34 (19 m, 15 f)		1632.16
	2d + 8–12w ^j^	16 (10 m, 6 f)		6607.31
P.I. ^g^	Pre Vacc	16 (10 m, 6 f)		1674.94
	1d + 8–12w	23 (13 m, 10 f)		4913.5
	2d + 8–12w	16 (10 m, 6 f)		9804.6

^a^ N = number participants; ^b^ % = percentage; ^c^ GMT = geometric mean titers; ^d^ m = male; ^e^ f = female; ^f^ Naïve = participants without previous SARS-CoV-2 infection; ^g^ P.I. = participants previously infected with SARS-CoV-2; ^h^ Pre Vacc = prevaccination; ^i^ 1d + 8–12w = 8–12 weeks after the first dose of vaccine; ^j^ 2d + 8–12w = 8–12 weeks after the second vaccine dose.

## Data Availability

All data available for this study are presented in the manuscript.

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
