# Peer review of "A Single Dose of ChAdOx1 nCoV-19 Vaccine Elicits High Antibody Responses in Individuals with Prior SARS-CoV-2 Infection Comparable to That of Two-Dose-Vaccinated, SARS-CoV-2-Infection-Naïve Individuals: A Longitudinal Study in Ethiopian Health Workers"

_vaccines, 2022, doi:10.3390/vaccines10060859_

Round 1
Reviewer 1 Report
The manuscript by Gelanew et al. reports a longitudinal study among Ethiopian healthcare workers who received CdAdOx1 vaccine against SARS-CoV-2. This study includes naïve, as well as previously infected (P.I.) individuals who received two doses of the vaccine and compares the levels of anti-RBD antibodies at baseline, after one and two vaccine doses. Albeit the sampled pools are quite limited, the authors convincingly demonstrate that previous infection contributes to a higher antibody titer at each time point. They also note that P.I. individuals after dose 1 display an ab titer similar to that possessed by naïve individuals after dose two. The authors rightly note that in countries with limited access to vaccine, some considerations (including theirs) would prompt to prioritize naïve individuals for the second dose. Yet, at least in my opinion, they correctly balance this consideration with the fast-mutating phenotype of SARS-CoV-2 (e.g. the recent advent of Omicron lineages), which prompts for the broadest diffusion of effective vaccines (2-3 doses) worldwide to reduce suffering of individuals and the burden on the health systems.
On overall, I deem this manuscript as reporting an earnest study pivoted around simple, yet interesting data, and I encourage the authors to correct a few minor issues to make it publishable.
Line 17: …longitudinal STUDY among…
Line 142: 87.5%, 21/23
Line 243: anti-RBD
Line 251: …this finding IS not…
Line 264: please space reference from the word “variants”
Author Response
We thank the reviewer for their time and effort in reviewing this manuscript. The reviewer's insightful and thorough comments and criticisms have greatly improved our work. Our point-by-point response to every comment is given below beneath each comment (colored).
The manuscript by Gelanew et al. reports a longitudinal study among Ethiopian healthcare workers who received CdAdOx1 vaccine against SARS-CoV-2. This study includes naïve, as well as previously infected (P.I.) individuals who received two doses of the vaccine and compares the levels of anti-RBD antibodies at baseline, after one and two vaccine doses. Albeit the sampled pools are quite limited, the authors convincingly demonstrate that previous infection contributes to a higher antibody titer at each time point. They also note that P.I. individuals after dose 1 display an ab titer similar to that possessed by naïve individuals after dose two. The authors rightly note that in countries with limited access to vaccines, some considerations (including theirs) would prompt to prioritize naïve individuals for the second dose. Yet, at least in my opinion, they correctly balance this consideration with the fast-mutating phenotype of SARS-CoV-2 (e.g., the recent advent of Omicron lineages), which prompts for the broadest diffusion of effective vaccines (2-3 doses) worldwide to reduce suffering of individuals and the burden on the health systems.
Response: Thank you for adequately highlighting the significance of our study and the implication/relevance of our findings in the context of the emergence of the new variants that can evade both natural infection- and vaccine-induced immune responses.
Overall, I deem this manuscript as reporting an earnest study pivoted around simple, yet interesting data, and I encourage the authors to correct a few minor issues to make it publishable.
Response: Again, we are grateful to the reviewer for making a favorable decision.
Line 17: …longitudinal STUDY among…
Response: It is now corrected. Please see line 17 in the revised version of the manuscript, which is marked in track changes.
Line 142: 87.5%, 21/23
Response: It is now corrected accordingly. Please see line 148 in the revised version of the manuscript, which is marked in track changes.
Line 243: anti-RBD
Response: It is now corrected. Please see line 246 in the revised version of the manuscript, which is marked in track changes.
Line 251: …this finding IS not…
Response: It is now corrected. Please see line 254 in the revised version of the manuscript which is marked in track changes.
Line 264: please space reference from the word “variants”
Response: Thank you. We have now amended it. Please see line 271 in the revised version of the manuscript marked in track changes.
Reviewer 2 Report
Dear Editor, despite the paper sounds interesting oa sething alreqdy proposed 9n other journala
Havervall S, Marking U, Greilert-Norin N, Ng H, Gordon M, Salomonsson AC, Hellström C, Pin E, Blom K, Mangsbo S, Phillipson M, Klingström J, Hober S, Nilsson P, Åberg M, Thålin C. Antibody responses after a single dose of ChAdOx1 nCoV-19 vaccine in healthcare workers previously infected with SARS-CoV-2. EBioMedicine. 2021 Aug;70:103523. doi: 10.1016/j.ebiom.2021.103523. Epub 2021 Aug 12. PMID: 34391088; PMCID: PMC8357428.
In light of small group i think that the paper shouls not be accepted
Author Response
We thank the reviewer for their time and effort in reviewing this manuscript. The reviewer's insightful and thorough comments and criticisms have greatly improved our work. Our point-by-point response to every comment is given below beneath each comment (colored).
Dear Editor, despite the paper sounds interesting oa sething alreqdy proposed 9n other journala
Havervall S, Marking U, Greilert-Norin N, Ng H, Gordon M, Salomonsson AC, Hellström C, Pin E, Blom K, Mangsbo S, Phillipson M, Klingström J, Hober S, Nilsson P, Åberg M, Thålin C. Antibody responses after a single dose of ChAdOx1 nCoV-19 vaccine in healthcare workers previously infected with SARS-CoV-2. EBioMedicine. 2021 Aug;70:103523. doi: 10.1016/j.ebiom.2021.103523. Epub 2021 Aug 12. PMID: 34391088; PMCID: PMC8357428.
In light of small group i think that the paper shouls not be accepted
Response. Although we and Harervall et al evaluated the antibody response of ChAdOx1 nCoV-19 vaccine in healthcare workers with and without prior SARS-CoV-2 infection, our study is different from theirs in the following key points.
- Firstly, studies that show the administration of a single dose adenovector (e.g., ChAdOx1 nCoV-19) vaccine induces equal or greater immune responses in individuals with prior SARS-CoV-2 infection are scarce. Moreover, the report by Harverall et al. is from a western country, Sweden. Additional studies from different geographical locations, races, and lifestyles are critical to understanding SARS-CoV-2 infection and/or vaccine-induced immunity. To the best of our knowledge, our report is the first from Ethiopia, if not from the continent of Africa. We believe in the importance of such studies from different parts of the world, especially Ethiopia, to make a more effective strategy for vaccine campaign policy decisions.
- Although the overall conclusion of our study is similar to that of Harverall et al., the composition of the two cohorts is different in terms of sex composition, i.e., the proportion of female participants in Harverall et al. was above 80%, whereas in our study was 42%.
- As reviewer #1 highlighted, we are not advocating the administration of single-dose vaccination for individuals with prior infection in the post-omicron wave considering this variant has been reported to evade both vaccine-induced and natural infection-induced immune responses.
Reviewer 3 Report
Review Report of Vaccines
Title: A single dose ChAdOx1 nCoV-19 vaccine elicits high antibody responses in individuals with prior SARS-CoV-2 infection comparable to that of double dose vaccinated SARS-CoV-2 infection naïve individuals: a longitudinal study in Ethiopian health workers.
Comments to the Authors:
- Authors have used single dose, double dose of COVID-19 vaccinated. Please analysis the results with the COVID-19 Booster dose.
- Author have conclude that, P. I. participants had significantly higher antibody titer compared to naïve participants, both after the first (GMT, 4913.50 24 vs. 1632.16) and second dose (GMT, 9804.60 vs. 6607.30). Please discuss the point more clearly.
- Authors should discuss: what are the side effects of COVID-19 vaccines?
- Discuss the SARS-CoV-2 infection in more details in the introduction section.
- Please polish the grammar.
- Please discuss the conclusion section of the article in more details.
I recommend it for publication after minor revision.
Author Response
We thank the reviewer for their time and effort in reviewing this manuscript. The reviewer's insightful and thorough comments and criticisms have greatly improved our work. Our point-by-point response to every comment is given below beneath each comment (colored).
Comments to the Authors:
1. Authors have used single dose, double dose of COVID-19 vaccinated. Please analysis the results with the COVID-19 Booster dose.
Response: Thank your comments and valuable suggestions. We agree with the reviewer on the importance of evaluating the impact of booster dose on the vaccine-induced immune response in both groups of our study participants. However, we are not in a position to do this for two main reasons. First, assessing this was not the scope of our study, as giving a third booster dose was not being advocated when we first designed the study. Second, to evaluate the impact of a third booster dose on antibody response, we need a third round of blood sample collection from study participants and prior to attempting the third round of blood sample collection following the booster dose, we are required to obtain approval from Ethics Review Committee after submitting the amended protocol. As such, we cannot be able to generate the required data in a short period of time.
2. Author have conclude that P. I. participants had significantly higher antibody titer compared to naïve participants, both after the first (GMT, 4913.50 vs. 1632.16) and second dose (GMT, 9804.60 vs. 6607.30). Please discuss the point more clearly.
Response: We thank the reviewer for this comment. Please see lines 237-240 which are marked in track changes in the revised version of the manuscript.
3. Authors should discuss: what are the side effects of COVID-19 vaccines?
Response: We noted relatively higher local pain at the site of injection and fever side effects within the period of 48 h following the first dose administration than following the second dose vaccination. In both cases, the adverse effects had been shown to resolve before 48 h suggesting the vaccine is tolerable. Although the main aim of this manuscript is to measure and compare vaccine-induced-antibody response between healthcare with and without prevaccination infection, we have now included this in the revised manuscript. Please see lines 260-262 which are marked in track changes.
4. Discuss the SARS-CoV-2 infection in more detail in the introduction section.
Response: We have now expanded the introduction about SARS-CoV-2 infection. See lines 31-36 of track changes in the revised manuscript.
5. Please polish the grammar.
Response: We have meticulously checked the grammar and made corrections accordingly as shown in track changes.
6. Please discuss the conclusion section of the article in more details.
Response: Thank you for this comment, we have now clearly elaborated the conclusion section. Please see lines 283-286 and lines 292-294, which are marked in track changes in the revised version of the manuscript.
I recommend it for publication after minor revision.
Response: We have incorporated the minor revision requested by the reviewer in our revised manuscript attached herewith and look forward to hearing the final acceptance of our manuscript for publication in the Vaccines Journal soon.
Round 2
Reviewer 2 Report
Paper could be accepted After that the authors modify It according to their statement about the unicity of the paper for african country. The should cite previous paper from havervall and discuss this in conclusion
Author Response
Comments and Suggestions for Authors
Paper could be accepted After that the authors modify It according to their statement about the unicity of the paper for african country. The should cite previous paper from havervall and discuss this in conclusion.
Response:
Thank you for the comment that we need to highlight the unicity of our work on ChAdOx1 nCoV-19 vaccine-induced immunity in the continent of Africa. We have now revised our manuscript as per your comment and inserted the paragraph below in our revised manuscript as shown in lines 3-273 highlighted in yellow. Also, we cited Harervall et al’s paper in the body of the revised manuscript and included it in the reference section as shown in lines 404-403.
“A prior study conducted by Harervall et al [32] among Swedish healthcare workers revealed that administration of a single dose ChAdOx1 nCoV-19 vaccine could induce equal or greater immune responses in individuals with prior SARS-CoV-2 infection. Additional studies from different geographical locations, races, and lifestyles are crucial to a better understanding of ChAdOx1 nCoV-19 vaccine-induced immunity. To the best of our knowledge, our study is the first from Ethiopia, if not from the continent of Africa, which provides a possible rationale for a single-dose vaccine regimen in countries with limited access to vaccines. Our findings from Ethiopia along with evidence from a western country, Sweden, support the superiority of “hybrid immunity” in eliciting a strong immunity against SARS-CoV-2 infection, comparable to that of two doses of the ChAdOx1 nCoV-19 vaccine in infection naïve individuals."